# Correlation of SARS-CoV-2 Neutralization with Antibody Levels in Vaccinated Individuals

**DOI:** 10.3390/v15030793

**Published:** 2023-03-21

**Authors:** Shazeda Haque Chowdhury, Sean Riley, Riley Mikolajczyk, Lauren Smith, Lakshmanan Suresh, Amy Jacobs

**Affiliations:** 1Department of Microbiology and Immunology, State University of New York at Buffalo, Buffalo, NY 14213, USA; 2Department of Oral Diagnostic Sciences, State University of New York at Buffalo, Buffalo, NY 14215, USA; 3KSL Diagnostics, Inc., Buffalo, NY 14225, USA

**Keywords:** SARS-CoV-2, COVID-19, antibody response, RNA viruses

## Abstract

Neutralizing antibody titers are an important measurement of the effectiveness of vaccination against SARS-CoV-2. Our laboratory has set out to further verify the functionality of these antibodies by measuring the neutralization capacity of patient samples against infectious SARS-CoV-2. Samples from patients from Western New York who had been vaccinated with the original Moderna and Pfizer vaccines (two doses) were tested for neutralization of both Delta (B.1.617.2) and Omicron (BA.5). Strong correlations between antibody levels and neutralization of the delta variant were attained; however, antibodies from the first two doses of the vaccines did not have good neutralization coverage of the subvariant omicron BA.5. Further studies are ongoing with local patient samples to determine correlation following updated booster administration.

## 1. Introduction

With over 102 million cases of and over 1 million deaths from COVID-19 worldwide since its identification in December 2019 (COVID Data Tracker, CDC), SARS-CoV-2 has become an increasingly persistent and strenuous challenge for healthcare professionals and the public alike. While COVID-19 is often characterized by upper respiratory symptoms including generalized malaise, fevers, nasal congestion, and cough, disease presentation is heterogeneous and can range from asymptomatic infection to multiorgan failure and death. Gastrointestinal symptoms include diarrhea, abdominal pain, nausea and vomiting, and anorexia [1], while acute myocardial injury, infarctions and other acute cardiac compromise can be presented by the patients [2]. Another complication of SARS-CoV-2 infection is a syndrome termed ‘Long COVID’, which is characterized by the persistence of diverse symptoms due to unidentifiable causes 12 weeks post-SARS-CoV-2 infection, and these have also proven to be a significant healthcare and economic burden [3].

As a beta-coronavirus, SARS-CoV-2 is closely related to SARS-CoV, as demonstrated through their structural proteins (envelope [E], membrane [M], and spike [S]), receptor (angiotensin-converting enzyme 2 [ACE2]), and receptor binding domains (RBD) on the S1 subunit of the spike protein [4]. Despite these similarities, the RBD in SARS-CoV-2 tends to be in the ‘lying-down’ conformation rather than the ‘standing-up’ conformation of the RBD in SARS-CoV, which may facilitate immune surveillance evasion by SARS-CoV-2, as demonstrated by the decreased development of neutralizing antibodies in patients with SARS-CoV-2 compared to SARS-CoV.

The patient-specific factors that have been demonstrated to increase risk for severe illness include increased age, obesity, tobacco use, and pre-existing comorbid conditions, such as hypertension, cardiovascular disease, and diabetes mellitus, though many of these are interrelated [5]. In addition to patient-specific factors associated with severe COVID-19 illness, disease manifestation is also dependent upon the molecular mechanisms of infection, such as the localization of the receptor, ACE2 and TMPRSS2, a serine protease which cleaves the spike protein into S1 and S2; co-expression in the gut, brain, kidney and cardiovascular system leads to viral entry and disease exacerbation [1].

The first vaccines were approved for emergency use in December 2019 by the U.S. Food and Drug Administration (FDA). These were mRNA-based vaccines, mRNA-1273 (Moderna) and BNT162b2 (Pfizer/BioNTech), which were also approved for use in Europe by the European Medicines Agency (EMA). Apart from these, non-replicating viral vector vaccines have also been in use; AZD 1222 (Oxford/AstraZeneca) has been in use in the UK, EU, and some Asian countries, and the AD26.COV2-S (Johnson & Johnson) has been approved for use in the US [6].

It has been widely demonstrated that low titers or neutralization potency of anti-RBD IgG antibodies are correlated with worsened morbidity and mortality outcomes [7]. This has been applied through the investigational use of convalescent plasma therapy in patients with impaired humoral immunity or in non-hospitalized patients who are at high risk for progression to severe disease under the Emergency Use Authorization (EUA) by the U.S. Food and Drug Administration (FDA). As the efficacy of this therapy is likely dependent on the infusion containing sufficient antibody levels, the EUA is for ‘high-titer plasma,’ which is defined according to a qualifying result that is specific for each of the bioassays and serologic binding assays accepted by the U.S. Food and Drug Administration (FDA). These bioassays are viral or pseudoviral neutralization assays and the serologic binding assays are either enzyme-linked immunosorbent assays (ELISA) or chemiluminescence assays (CLIA).

While previous studies have demonstrated a correlation between anti-RBD IgG levels and neutralizing antibody titers, these were often conducted using ELISA and microneutralization assays. To our knowledge, there are no prior studies that use CLIA and plaque reduction neutralization (PRNT) assays, both of which have been shown to be more sensitive than their respective counterparts, to elucidate this relationship. Herein, we show the correlation between CLIA and PRNT assay results for patients who received the primary doses of the mRNA-1273 (Moderna) and BNT162b2 (Pfizer/BioNTech) vaccines. All samples were tested for the neutralization capability of both Delta (B.1.617.2) and Omicron (BA.5).

## 2. Materials and Methods

Chemiluminescence Immunoassays—The detection of antibodies to the RBD on S1 was performed on serum at the KSL Diagnostics Laboratories using KSL chemiluminescence immunoassays (CLIA). In this technique, electromagnetic radiation caused by a chemical reaction produces light. The analytic reaction produces visible or near-visible radiation generated when an electron transitions from an excited state to ground state. The luminophore markers used in KSL CLIA assays are acridinium esters. The CLIA assay provides qualitative detection of human IgA, IgG, or IgM antibodies to SARS-CoV-2. The test was authorized by NY State and EUA on 25 November 2020, under Project ID No. 84341. The samples were incubated with a magnetic bead coated with the S1 subdomain from SARS-CoV-2. Once the unbound materials were washed away by magnetic separation, the acridinium ester marker is added for incubation. After a wash step, SARS-CoV-2 antibodies are detected with a substrate that produces a luminescence reaction with the acridinium ester. The luminescence intensity of acridinium ester is proportionate to the amount of antibody against novel coronavirus and yields a test result expressed by cut-off index (COI). If the sample value is less than 0.8 COI, no SARS-CoV-2 antibody is detected. If the value is within 0.8–1.0 COI, the SARS-CoV-2 antibody is indeterminate. If the value is greater than 1.0 COI, then SARS-CoV-2 antibody is detected.

Virus Stock Generation—Inhibition of infection by the test sera was studied by infecting the sera with the viral variants, SARS-CoV-2 Delta (B.1.617.2) and Omicron (BA.5). Virus was obtained from BEI Resources were propagated in Calu-3 (ATCC, VA) for 3 days (Delta) or 7 days (Omicron). Infection was confirmed by the observance of cytopathic effect (CPE), such as the rounding and dislodgement of cells. Harvested virus was then titered using a PRNT assay so that a viral load of 6 × 10^3^ PFU/mL (30 PFU/well) could be added to the test samples.

Plaque Reduction Neutralization Assays—The PRNT50 protocol used was adapted from Bewley et al [8]. Briefly, serum samples were diluted in a 96-well plate, before SARS-CoV-2 virus was added to the diluted serum at BSL-3 and neutralization was allowed to occur. The neutralized or control virus was then transferred onto Vero-E6 cells (ATCC, VA), allowed to adsorb, overlaid with 1% agar in DMEM and incubated at 37 °C and 5% CO_2_. The incubation time for viral propagation and infection was increased from 3 days to 4 days when using the Omicron variant as compared to the Delta strain, since plaques formed by the omicron variant were smaller and took longer to develop. Plates were then fixed with 4% paraformaldehyde, and agar plugs were removed so that plaques could be counted and scored (Figure 1). PRNT50 values were then calculated using Prism GraphPad (Version 9.5.0) according to the instructions outlined in Bewley et al. (Figure 2). Samples were tested in two biological and two technical replicates.

Sample characteristics—In this study, 110 participants (males or females) between the ages of 18 and 85 with no known prior SARS-CoV-2 infection as confirmed by RT-PCR were included. The presence of SARS-CoV-2 nucleocapsid antibodies was considered an exclusion criterion. Participant inclusion and exclusion criteria are listed in Table 1. Participants had received the first 2 doses of the mRNA-1273 (Moderna) (55 patients) and BNT162b2 (Pfizer/BioNTech) (55 patients) vaccine as per the provider’s protocols. Fifteen days post vaccination, sera were tested for IgG levels using KSL Chemiluminescence Immunoassays (CLIA), and samples were grouped according to the amount of IgG measured (Table 2). Following that, a total of 17 samples from all the groups were tested for viral neutralization using live virus in BSL-3.

Statistical analysis—Pearson’s correlation test was used to assess the correlation between calculated PRNT50 titers and IgG COIs or age. *p*-values ≤ 0.05 were considered statistically significant. The neutralization induced by either vaccine was compared using a boxplot displaying median and 95% CI. Data analysis and visualization was conducted in Prism GraphPad (Version 9.5.0).

## 3. Results

### 3.1. IgG COI Values Correlate with PRNT50 Titers

It was observed that the calculated PRNT50 values correlated with the measured IgG COI (Figure 3), with the delta variant showing a stronger statistically significant correlation (r^2^ = 0.6908, *p*-value < 0.0001) compared to the omicron variant (r^2^ = 0.165, *p*-value = 0.0756 n.s.). At COIs < 20.0, no neutralization was observed, with the number of plaques counted being the same as the virus only control (VOC) of both viral strains tested. As summarized in Table 3, when tested with delta variant, robust neutralization was observed at COIs >20.0, as high titers were obtained, while suboptimal neutralization and low titers were obtained for COI between 10.0 and 20.0. In contrast, when tested with the omicron BA.5 variant, suboptimal neutralization indicated by low titers was observed at COIs >10.0.

### 3.2. Age Does Not Correlate with PRNT50 Titers

Analysis showed that the age of the patient during vaccination did not have any effect on the PRNT50 titers obtained (Delta r^2^ = 0.015, *p*-value = 0.6332 n.s.; Omicron r^2^ = 0.041, *p*-value = 0.4348 n.s.) (Figure 4). This was in tandem with the result that the measured IgG COIs also did not correlate with age (r^2^ = 0.0343, *p*-value = 0.4620 n.s.).

### 3.3. Comparison of Neutralization following Primary Doses of Either the Moderna or Pfizer/BioNTech Vaccines

The average median PRNT50 titer obtained following vaccination with the primary doses of the Pfizer vaccine was 242.2 when challenged with the Delta strain, and 146.1 when challenged with the Omicron strain. In comparison, following primary doses of the Moderna vaccine, the average median titer when challenged with the Delta strain was 78.31, while it was 118.2 when challenged with the omicron strain (Figure 5). This indicates that neutralization capacity by antibodies produced following vaccination with the Pfizer mRNA vaccine was more effective when compared to the neutralization by antibodies produced by the Moderna vaccine—a result that held true when challenged with both the delta and omicron strain.

## 4. Discussion

Humoral immunity is characterized by the production of antibodies by B cells as a response to antigens. Although both IgM and IgA appear within the first week of symptom onset, IgG is the most abundant antibody type and provides longer-lasting immunity. IgG is seen in circulation from about 7 days onwards [9]. This response of immunity is also typical for SARS-CoV-2. IgG titers remain stable for at least 4 to 6 months following diagnosis among PCR-confirmed individuals, whereas IgA and IgM titers rapidly decay. Antibodies targeting the spike glycoprotein of the SARS-CoV-2, especially the receptor binding domain (RBD) within the S1 subunit, show the highest neutralizing capacity. The presence of neutralizing antibodies is considered a functional correlate of immunity and provides at least partial resistance to subsequent. Although some serological assays showed a high correlation between IgG and neutralizing antibodies [10], others have poor correlation [11]. Therefore, comparison with virus-neutralization experiments is important as part of the validation of new serological assays.

Several laboratory-developed and commercially available assays utilizing various technology platforms are available to detect anti-SARS-CoV-2 spike antibodies. While these platforms provide a high-throughput means of detecting antibodies against SARS-CoV-2, they are unable to measure the immunological function of SARS-CoV-2-specific antibodies. In contrast, the plaque-reduction neutralization test (PRNT) quantifies levels of neutralizing antibodies capable of blocking the interaction that mediates virus entry into susceptible host cells and subsequent virus replication [12]. For SARS CoV-2, this interaction involves binding of the RBD of the spike glycoprotein with the ACE2 on host cells. This makes the conventional PRNT the reference standard for the evaluation of virus-neutralizing antibodies. Prior to this study, to our knowledge, no other study has compared the effect of neutralizing antibodies on the omicron or delta strain using the plaque reduction neutralization test, although a few have studied the effects of the USA-WA1/2020 isolate [13].

In our lab, testing the neutralizing capabilities of vaccine-induced IgG antibodies showed that at very low IgG levels, no neutralization is achieved. Further, there is a positive correlation between the measured IgG using CLIA and PRNT50 values obtained. As expected, neutralization of virus by vaccine-induced antibodies is more robust when challenged with the delta strain when compared to the omicron BA.5 strain. This is similar to what is reported in the literature [14]. Since the samples obtained are from individuals vaccinated with only the first doses of the vaccine, but not the boosters, the antibodies produced are not effective against the omicron variant. Studies have shown that neutralizing antibodies (nAbs) are progressively less effective against each new variant of concern (VOC) or variant of interest (VOI). This is especially true for omicron BA.5 subvariant, used in this study, which escapes nAbs due to extensive mutations and antigenic remodeling of its spike trimer, and predominance of the closed state of the spike protein [7]. Compared to the original strain, the omicron BA.1 variant has 35 mutations, 15 of them in the RBD which is the region that binds to the host cell receptors and is a target of nAbs. Of the 15 mutations, nine fall in the region specific for binding to the ACE2 receptor on host cells, thereby allowing stronger binding and nAb escape [15]. The subsequent variants that emerged, BA.2, BA.2.12.1, BA.4 and BA.5, each have more numerous and unique mutations, which make the circulating omicron variants more transmissible and less susceptible to vaccine induced immunity. In fact, it has been determined that efficiency of the Moderna vaccine following the first booster (third dose) is only high against the omicron BA.1 variant, and the second booster (fourth dose) is required to achieve a high vaccine efficiency against BA.2, BA.2.12.1, and BA.4, while vaccine efficiency against BA.5 is low even after the fourth dose [16]. The same was also observed when testing vaccine efficiency of the Pfizer/BioNTech vaccine [17]. Further, even though the fourth dose conferred high vaccine efficiency against the BA.5 variant, the effect waned within six months in both studies.

However, it has been reported that some classes of neutralizing antibodies which exert their function by binding to the less immunologically challenged and therefore more conserved stem-helix region of the S2 domain show neutralization at very high concentrations (IC50 > 1 ug/mL) in vitro while in in vivo studies, they show the same effect at lower concentrations [18,19]. Further, it is important to note that in the in vitro studies conducted in our lab, it was observed that infectivity of omicron was to a lesser degree when compared to the delta strain. This was concluded due to the longer infection cycles required to obtain plaques when infecting with omicron. Further, titering of viral particles showed lower numbers of plaques for dilutions of omicron when compared to the same dilution series of the delta variant. This might attribute to the less severe disease manifestation during omicron infections [20]. However, further studies are required to assess this observation.

It was also observed that the age of the patient did not correlate with either the IgG COIs measured, or the PRNT50 titers obtained. However, it has been previously shown that following vaccination, age plays a significant role in neutralization of SARS-CoV-2 by vaccine-induced antibodies, wherein an increase in age correlates with decreased immunity [21]. The results obtained in our study can be attributed to a small sample size and a limited number of participants of different ages in each IgG COI group. It must also be noted that certain individuals who received the primary doses of the vaccine did not produce any measurable IgG, and this may be attributed to an impaired immune system of the patient.

In our study, we observed that vaccination with the primary doses of the BNT162b2 (Pfizer/BioNTech) vaccine induced antibodies that produced higher PRNT50 titers and thereby caused more effective neutralization when compared to the effect of the mRNA-1273 (Moderna) vaccine, which led to lower PRNT50 titers overall, specifically against the delta strain; the effects on omicron were substantially low and similar. It was previously shown that the Moderna vaccine induced a higher amount of functional antibodies; the difference observed in our study is relatively small, and can be attributed to a small sample size [22].

While assays such as ELISA measure the amount of antibodies present, the PRNT provides an insight into the neutralizing functionality of the circulating antibodies. This makes it the gold standard for assessing the neutralizing capability of antibodies. However, even though the PRNT is often used as the reference standard for the evaluation of virus-neutralizing antibodies, this assay is time-consuming, laborious, and requires biosafety containment level 3 (BSL-3) facilities to work with the high-risk group-3 pathogen. As such, this is not practical for large-scale community testing, due to low turnaround time and high manual input. In this study, due to the high labor demand and low efficiency of testing method, a small sample size was studied, which is a limitation of the study, and which may attribute to the differences observed between existing literature and our observations. However, using the tested samples as a reference, high throughput testing techniques can be designed and validated. One such technique is a microneutralization assay (currently being validated in-house) which uses labeled antibodies to count plaques in a 96-well or 384-well format [8]. This provides an added advantage over other immunofluorescence techniques such as ELISA because it counts the foci of infected cells instead of just the absorbance. Future studies will be aimed at assessing the neutralizing capabilities following vaccinations with the booster shots, and also characterize the types of IgG and their effect on neutralization. Trends in variant mutation of SARS-CoV-2 and their nAb escape make it important to develop efficient and high-throughput testing capability to analyze large data sets and validate community-based conclusions.

## 5. Conclusions

The amount of IgG produced following vaccination with the primary doses of the mRNA-1273 (Moderna) and BNT162b2 (Pfizer/BioNTech) vaccines correlates positively with the PRNT50 titers obtained when challenged with the SARS-CoV-2 Delta (B.1.617.2) variant, although this was not observed when challenged with the Omicron (BA.5) variant, meaning that immunity offered by antibodies produced by the first vaccine doses is effective at neutralizing the delta variant only. No significant correlation was observed with age and no significant difference was observed between the two vaccines administered. The conclusions of this work corroborate the importance of fine-tuning the vaccines to the current strains that dominate in the population and potentially tailoring to individual regions as well.

## Figures and Tables

**Figure 1 viruses-15-00793-f001:**
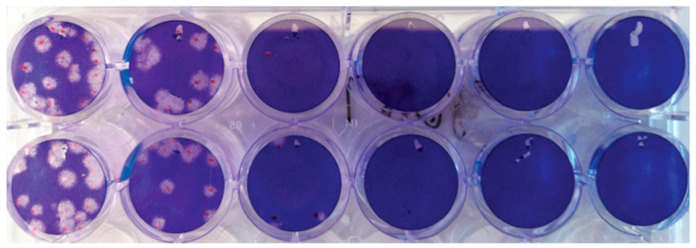
Example of plaque reduction neutralization performed in duplicate. Overlay fixed with 4% paraformaldehyde and stained with crystal violet before plaques were counted.

**Figure 2 viruses-15-00793-f002:**
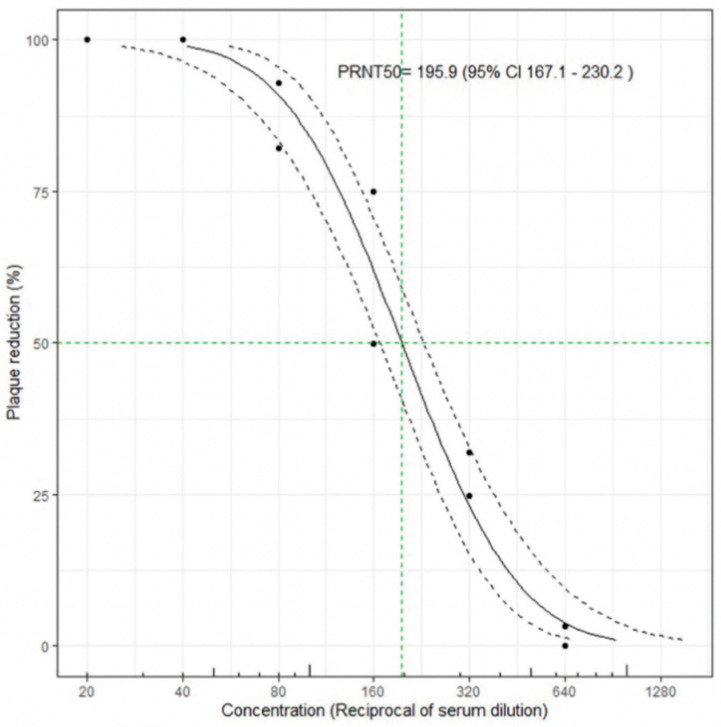
Representative PRNT50 analysis performed according to Bewley et al. [8].

**Figure 3 viruses-15-00793-f003:**
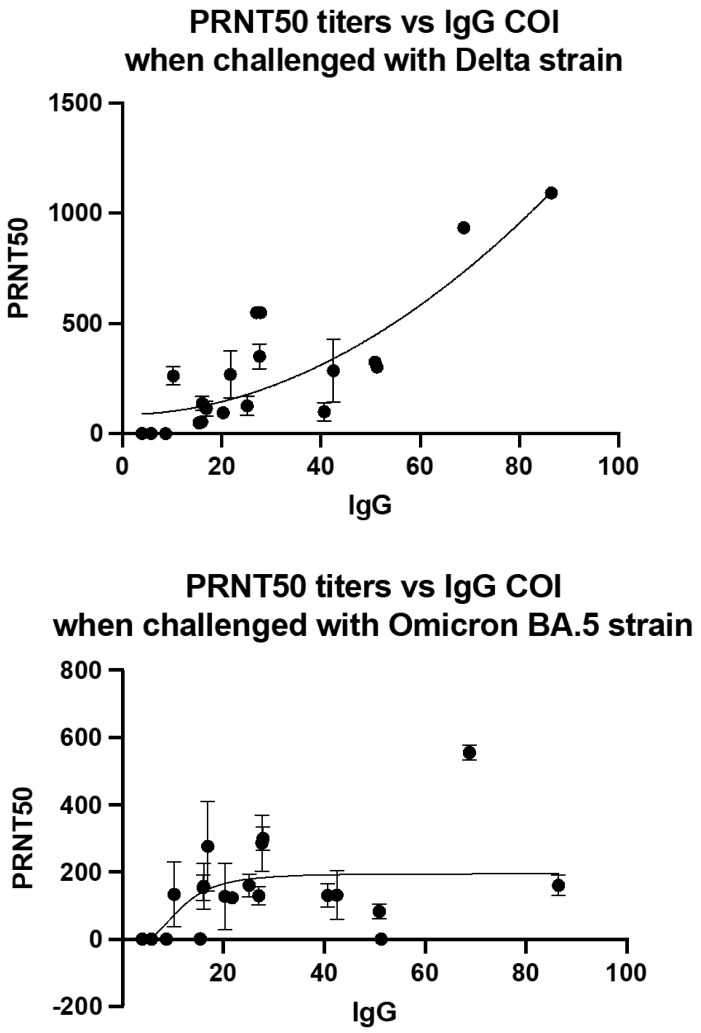
Correlation of PRNT50 titers with IgG COI when challenged with Delta and Omicron variants of SARS-CoV-2. Viral load of 6 × 10^3^ PFU/mL (30 plaques/well) were added to Vero-E6 cells. Data represented mean ± SD of two biological replicates. µ = 0.05.

**Figure 4 viruses-15-00793-f004:**
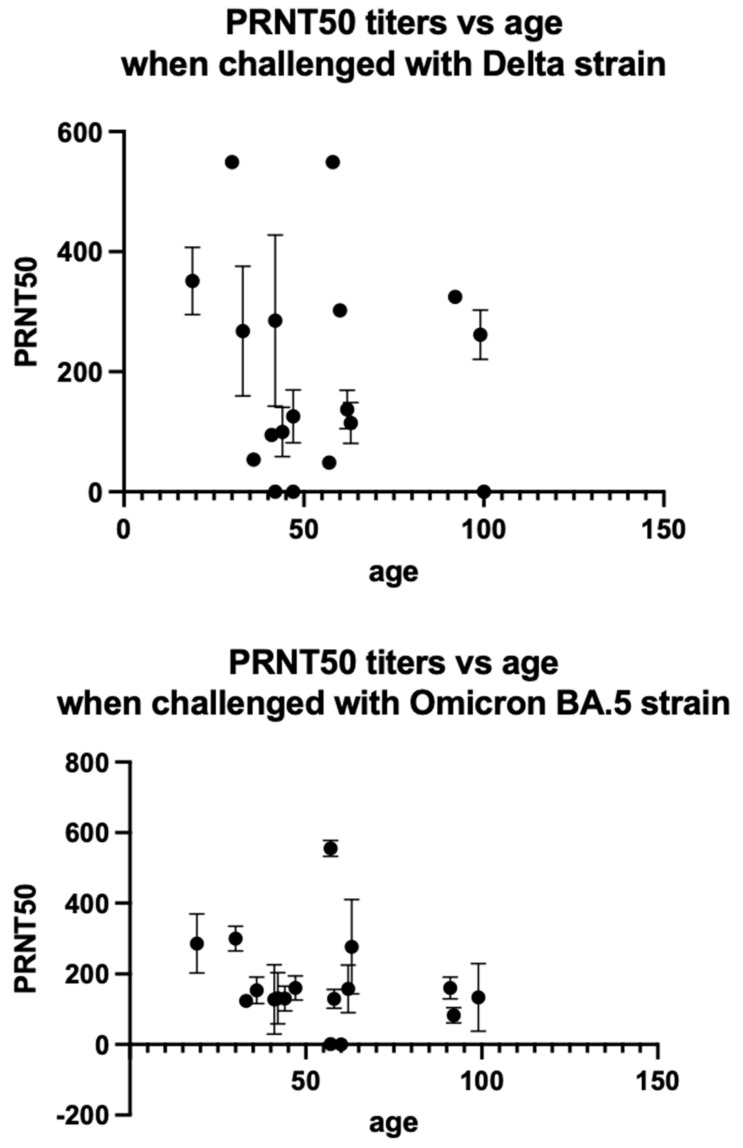
Correlation of PRNT50 titers with age when challenged with Delta and Omicron variants of SARS-CoV-2. Viral load of 6 × 10^3^ PFU/mL (30 plaques/well) were added to Vero-E6 cells. Data represented mean ± SD of two biological replicates. µ = 0.05.

**Figure 5 viruses-15-00793-f005:**
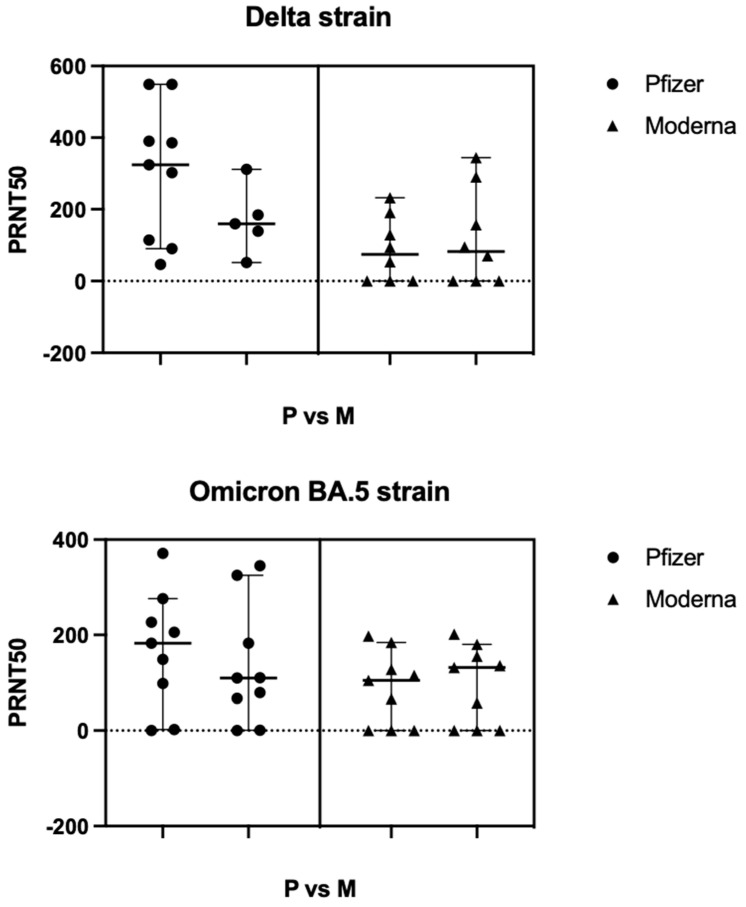
Comparison of PRNT50 titers obtained following vaccination with the primary doses of the Moderna or Pfizer/BioNTech mRNA vaccines. Viral load of 6 × 10^3^ PFU/mL (30 plaques/well) were added to Vero-E6 cells. Data represented median ± 95% CI of two biological replicates.

**Table 1 viruses-15-00793-t001:** Inclusion/Exclusion Criteria of participants enrolled in study.

**Inclusion Criteria**
1. Age and Sex: Male or female participants between the ages of 18 and 85 at the time of enrollment
2. Participants with no known exposure to COVID-19 infection
3. Had FDA approved vaccines administered as per the vaccine company recommended protocols
4. Participants who are willing and able to comply with all scheduled visits and laboratory tests.
Exclusion Criteria
1. Age below 18 years of age
2. History of COVID-19 infections
3. History of antibodies to SARS-CoV2 nucleocapsid antibodies
4. People with non-adherence to vaccine administration protocols

**Table 2 viruses-15-00793-t002:** Sample groups according to IgG COI.

Group	COI * Value Range
1	<0.5
2	0.5–<1.0
3	1.0–2.99
4	3.0–5.99
5	6.0–9.99
6	10.0–14.99
7	15.0–19.99
8	20.0–24.99
9	25.0–29.99
10	30.0–39.99
11	40.0+

*—Cut-off Index.

**Table 3 viruses-15-00793-t003:** Average PRNT50 values obtained for different COI groups when challenged with the delta or omicron variant.

Group Number	COI Value Range	PRNT50 Average (Delta)	PRNT50 Average (Omicron)	Conclusion
1–5	0–9.99	Same as VOC	Same as VOC	No neutralization of both strains
6–7	10–19.99	123.14	144.84	Similar neutralization of both strains
8–10	20–39.99	322.92	187.81	Delta neutralized more effectively than Omicron
11	>40	235.06	176.58	Delta neutralized more effectively than Omicron

## Data Availability

The data presented in this study are available on request from the corresponding author.

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
