# Peer review of "Correlation of SARS-CoV-2 Neutralization with Antibody Levels in Vaccinated Individuals"

_viruses, 2023, doi:10.3390/v15030793_

Round 1
Reviewer 1 Report
It is quite an interesting manuscript. The topic of this manuscript falls within the scope of Viruses Journal.
The Authors have presented sufficient data. The appropriate tables and figures have been provided. The article is easy to read and logically structured. The authors used appropriate statistical methods.
There are some comments in the reviewer's opinion that should be taken under consideration by the Author:
1. In the introduction section please cite more papers f.e.
Doi: 10.3390/jcm10214802
doi: 10.1161/CIRCRESAHA.121.317997.
doi: 10.1080/07853890.2022.2076901.
2. In the introduction section please give some information about which vaccines are most commonly used in Europe and USA.
doi: 10.3390/jcm11030750
doi: 10.5582/bst.2021.01061
DOI: 10.1186/s12889-021-10862-1
3. In the method section please add: what were the inclusion and exclusion criteria of patients?
4. In the method section please give more data on patients: age, BMI, sex etc.
5. Please more preciously give the limitations of your study
6. Please give the conclusion sections
7. Some references are missing all data
Author Response
- Introduction expanded to include more details about gastrointestinal and cardiovascular symptoms, and long covid.
- Information about vaccines approved and in use in the US and EU included in the introduction.
- Table 1 included to list inclusion and exclusion criteria of participants.
- Ages (18-85) and sex (M/F) of sample included. Other information requested not available from the agency providing samples.
- On page 9, "In this study, due to the high labor demand and low efficiency of testing method, a small sample size was studied, which is a limitation of the study, and which may attribute to the differences observed between existing literature and our observation." included to address limitation.
- Conclusion section included after discussion.
- Reference list revised to ensure all data included.
Reviewer 2 Report
This study provided useful information and the manuscript is well-written. I just have two minor suggestions.
1. Pleae add one table to describe the demographic features of the included 110 patients.
2. Please shorten the first two paragraph in the discussion and focused on discussion your finding rather than literaure review.
Author Response
- Age and sex of sample included in an added table 1 as other demographic features not available.
- Revisions made to discussion to the extent possible without losing relevant information.
Reviewer 3 Report
Interesting and relevant topic is discussed, but some revisions are needed.
The title of the article “Relationship and Risk Stratification of COVID 19 Vaccinated Individuals based on Antibody Levels’’ does not fully match the content, in my opinion.
There must be Section Materials and Methods, not only Methods.
Materials such as Viruses and Sera are not described well. It was not clear whether the patients from whom the sera were taken were infected or vaccinated. What is the origin of viruses used in PRNT?
It is better, if the methods and materials are presented in subsections.
Section Discussion. First paragraph, on page 8, line 7 has a large space.
The text in this section needs revision and clarification. It is not arranged well. There is a lot of information, but it is not well connected to the results. The second paragraph, page 8 should be split into two. There is no clear conclusion.
Author Response
A subsection added within materials and methods to describe where virus was obtained from, and how virus stocks were generated. Information included in acknowledgements to cite virus source (BEI resources).
Table 1 included to explain participant characteristics more clearly (demographics, inclusion/exclusion criteria).
Title modified to "Relationship of SARS-CoV-2 Neutralization with Antibody levels in Vaccinated Individuals" to better reflect content.
Conclusion section added to provide a more focused view.